# The journey to diagnosis and care of functional neurological disorder (FND)

Ivet Pritomanova[1], Sarah R. Cope[1,2], Billie James[3], Mark J. Edwards[4], Jared G. Smith[1,5], Sharif El-Leithy[6], Serena Vanzan[1], Patricia Hogwood[7], Dawn Golder[7], Kati Jane Turner[7], Jo Billings[8]*

1 Clinical Research Unit, South West London and St George's Mental Health Trust, Springfield University Hospital, London, England, United Kingdom, 2 Department for Clinical Neuropsychology and Clinical Health Psychology, St George's University Hospitals Foundation NHS Trust, London, England, United Kingdom, 3 Surrey and Borders Partnership NHS Foundation Trust, England, United Kingdom, 4 Department of Basic and Clinical Neuroscience, Institute of Psychiatry, Psychology and Neuroscience, King's College London, London, England, United Kingdom, 5 Global, Public and Population Health and Policy, St George's School of Health and Medical Sciences, City St George's, University of London, London, England, United Kingdom, 6 Traumatic Stress Service, South West London and St. George's Mental Health NHS Trust, Springfield University Hospital, London, England, United Kingdom, 7 London, United Kingdom, 8 Department of Psychiatry, University College London, London, England, United Kingdom

* j.billings@ucl.ac.uk

## Abstract

### Purpose

Functional neurological disorder (FND) is a common condition, associated with high disability and healthcare costs, and poor treatment access. This qualitative study aimed to explore participants' experiences of being diagnosed with functional neurological disorder (FND), accessing treatment, and navigating life after diagnosis. Participants were drawn from a randomised feasibility study of eye movement desensitisation and reprocessing therapy for people with FND (MODIFI, Trial Registration: NCT05455450 (www.clinicaltrials.gov)).

### Methods

Reflexive thematic analysis was used to analyse the data from eighteen semi-structured interviews with participants diagnosed with FND.

### Results

Six main themes were identified: (1) the process of seeking a diagnosis and initial relief from the physical struggles, (2) receiving a diagnosis of FND, (3) treatment for FND, (4) the burden of FND in day-to-day life, (5) the need to assert agency and return to normal, and (6) hopes for the future.

**Data availability statement:** Data in this study (interview transcripts) contain potentially identifying or sensitive patient information and has not been made publicly available, in line with ethical approvals for this study. Requests for further information can be made to NHS West Midlands Edgbaston Research Ethics Committee and Health Research Authority (ref: 22/WM/0178) at edgbaston.rec@hra.nhs.uk.

**Funding:** This project is funded by the National Institute for Health and Care Research (NIHR) under its Research for Patient Benefit (RfPB) Programme (Grant Reference Number 202277). The funders had no role in study design, data collection and analysis, decision to publish, or preparation of the manuscript.

**Competing interests:** The authors have declared that no competing interests exist.

## Conclusion

The findings emphasize the need for personalised and compassionate care for individuals suffering from FND, underpinned by increased service provision within the healthcare system.

## Introduction

Functional neurological disorder (FND) is a common condition characterised by neurological symptoms (e.g., weakness, tremors, seizures) which appear to result from a loss of voluntary control over movement or perceptual experience, despite normal structure of the nervous system. Precipitating factors for developing FND can include injuries and psychological trauma [1,2]. Experiencing adverse life events is eight times more common in people with FND, although not all people with FND will have experienced significant trauma [3].

It can take a long time for patients to receive a correct diagnosis of FND [4]. In a large randomised controlled trial (RCT) for functional (dissociative) seizures (CODES), participants reported a mean of six years between FND symptom onset and receiving a diagnosis [5]. Before receiving an FND diagnosis, it is not uncommon for some people to face misdiagnosis with other neurological disorders, such as epilepsy and multiple sclerosis, or other psychiatric conditions [6]. Even though there can be delays to diagnosis, FND has a low misdiagnosis rate [6–8]. Following diagnosis, patients can face stigma due to healthcare professionals' (HCPs) limited knowledge of FND and negative attitudes [9]. For example, a recent study looking at British healthcare professionals' attitudes towards FND highlighted that HCPs displayed strong subconscious biases that FND is illegitimate in comparison with other conditions such as multiple sclerosis [10]. Patients' lived experiences of FND illustrate significant disability and distress and the need for appropriate treatment and management [11].

A recent systematic review estimated a minimum point prevalence between 80–140/100,000. Therefore, FND is more prevalent than other common neurological disorders, e.g., multiple sclerosis has a period prevalence rate worldwide of 30/100,000 [12]. Treatments for FND are applied differentially across patient populations and can include education, inpatient treatment, physiotherapy, and psychological therapy, but only some interventions have been formally trialled [13–17] and access to treatment is limited. Some patients are not offered any treatment, whilst others can receive unhelpful care, or even harmful treatment due to misdiagnosis [18]. The path towards obtaining an FND diagnosis and appropriate treatment can be both ambiguous and time-consuming. There has been limited investigation into the lived experiences of life with FND. Our aim was to shine a light onto patients' journeys of symptoms-to-diagnosis, and explore how they live with their symptoms after diagnosis. Previous qualitative accounts have explored similar areas, but using different methodologies [11,19].

This qualitative study was nested within the MODIFI randomised controlled trial (RCT) evaluating the feasibility of using eye movement desensitisation and

reprocessing therapy (EMDR) for people with FND [20]. MODIFI was a single-blind RCT with two arms: EMDR (plus standard neuropsychiatric care (NPC)) and NPC [21]. Participants randomised to EMDR+NPC were offered up to sixteen EMDR sessions, and standard outpatient neuropsychiatric appointments (NPC) [22]. NPC was standard care and included 1–3 routine out-patient appointments with a neuropsychiatrist during the trial period, as well as invitation to routine educational group sessions run in the service. The MODIFI trial ran from 01/06/2022–30/05/2025. Data for the qualitative interviews described in this paper was collected between 1st December 2023 and 30th July 2024.

As part of the trial, a sample of participants from each arm were invited to take part in a semi-structured interview to explore their experiences of the trial and their FND journey.

## Materials and methods

### Reflexivity

The core team involved in this qualitative study were women from ethnically diverse backgrounds and various occupations and included experts by experience. A wider group of researchers drawn from the MODIFI trial team, including men, contributed to the final paper. The first author, IP, holds a BSc in Psychology and has experience as a research assistant in qualitative research, including work on analysis via reflexive thematic analysis, in the field of health and clinical psychology. The main supervisor, JB, is a Consultant Clinical Psychologist and Professor of Psychological Trauma and Workplace Mental Health. The Chief Investigator of the clinical trial, SRC, is a Consultant Clinical Psychologist with expertise in FND. The Clinical Trial Manager, SV, coordinated the qualitative study. A Patient and Public Involvement (PPI) panel – KJT, PH, DG – consisted of three women with lived experience of FND, one of whom is a qualitative researcher. The PPI representatives were closely involved in the design of this study, developing the interview schedule (please see supporting information S1 File) and analysing the results.

As a team, we brought a diversity of experience and perspectives to this study. The degree to which we had differing previous experience of FND may have shaped our understanding and interpretation of the interview data, which we sought to balance by regular research team discussion and reflection. The first author, IP, utilised a reflexive diary whilst analysing this data, to note down personal thoughts and feelings. Examples were concerns about presenting the data in the best way, ensuring that voices of participants were not silenced, and reflecting upon a realisation of the privilege to work with this dataset. These reflections were discussed and explored by the supervisory team alongside evolving analysis of the interview data.

### Methodology

Participants who had completed their time in the MODIFI trial were invited to take part in an interview by e-mail and text message. Following verbal and written consent, interviews were carried out either online using Microsoft Teams, or at the hospital site, with a research assistant. Payment of £20 was given following participation. The interview was split into two parts: the first part explored the journey of participants and their history prior to enrolling in MODIFI, focusing on FND diagnosis and treatment. The interview guide was developed by the research team in consultation with our PPI group (please see supporting information S1 File). This paper will describe the findings of this part of the interview. The second part of the interviews that covered participants' experiences of the trial is reported separately.

The interviews were transcribed verbatim using the Microsoft Teams transcription function and this was checked and corrected by BJ prior to data analysis. Participants were pseudo-anonymised by using participant IDs and removing any information that could potentially identify them. The interviews were analysed using reflexive thematic analysis facilitated by NVivo 14. We followed the five-step model for conducting reflexive thematic analysis by Braun and Clarke [23,24]. First, IP listened to the participant interviews and read the transcripts, sharing five transcripts with the PPI representatives (PH, DG, KT) for data familiarisation. Having initially coded the five transcripts on paper to utilise creative methods, IP coded the rest of the transcripts using NVivo 14. As the third step, IP used an inductive approach to generate the initial

themes from the identified codes in NVivo and produced a thematic table, focusing on the most relevant themes, sub-themes and quotes extracted from the dataset; JB reviewed the thematic table and provided suggestions which IP incorporated. Then, IP presented the thematic table to the wider research team and the team further refined the initial themes. Finally, IP defined and shaped the final themes. The entire team reviewed the final version of the analysis. PPI involvement and engagement were incorporated throughout the research design and analytical stages to enhance robustness. We have kept a clear audit trail of our analysis and decision making throughout the study.

## Participants

MODIFI recruited 50 participants, randomised to two arms: EMDR+NPC, and NPC alone. Recruited participants had to have a diagnosis of functional seizures and/or functional motor symptoms to be recruited, and some also experienced additional functional neurological symptoms. Participants chose a maximum of two most disabling symptoms at the beginning of the trial period to rate, e.g., seizures, tremor, limb weakness, tingling/numbness, gait disturbance. Of the 25 participants randomised to the EMDR+NPC arm, 23 reached the end of the trial and were invited to take part in the interview. Out of these, 11 consented to take part and were interviewed. Of the 25 participants randomised to the NPC only arm, 15 were invited using purposive sampling to ensure a balanced range of symptom presentations. Of the 15 invited, 7 consented and were interviewed. In total, 18 participants took part in semi-structured interviews between December 2023 and July 2024, lasting on average 55 minutes (range 25–90 min), with the majority held online (online = 16; in-person = 2). No adverse effects during the interviews were recorded. Fourteen participants identified as female, one as male, three preferred to self-describe or not disclose their gender. The mean age for the sample was approximately thirty-three years (range 18–66). All but four participants in the sample identified ethnically as White. The participants predominately experienced more than one category of functional neurological symptoms. Mean time since FND symptoms started as 3 years (range: 1–12). Mean time since FND diagnosis was 1 year (range 0–6). (See Table 1 for the participant characteristics).

## Ethics

This study was part of the MODIFI RCT which was reviewed by the NHS West Midlands Edgbaston Research Ethics Committee and Health Research Authority (ref: 22/WM/0178) and approved on 27/09/2022 (IRAS ID: 311719) http://dx.doi.org/10.17504/protocols.io[10.1136/bmjopen-2023-073727]. All participants provided written and oral informed consent. The study was conducted in accordance with the principles of the Declaration of Helsinki. Trial Registration: NCT05455450 (www.clinicaltrials.gov).

# Results

Through inductive and iterative thematic analysis of the data we identified six main themes and 13 sub-themes, presented in a thematic table in Table 2.

## Theme 1. Is FND real?

This theme captures participants' experiences of their initial symptoms of FND and seeking help. Almost uniformly, patients faced stigma, lack of understanding and awareness about FND from HCPs. Participants shared that they felt they were not being listened to during their initial interactions with HCPs, with a few participants adding that the people in their life, such as work colleagues, also did not acknowledge the struggles of dealing with FND.

   **1.1.  Experiences of the onset of FND.**  Participants described the variety of symptoms they experienced at the onset of FND, such as seizures, tremors, balance issues, fatigue, dissociation, passing out, dizziness and others. Some participants doubted the reality of what was happening to them.

   One participant noted that they downplayed their symptom presentation:

**Table 1. Participants' characteristics.**

|  | N (Total N = 18) |
|---|---|
| **Gender** |  |
| Female | 14 |
| Male | 1 |
| Prefer not to say | 1 |
| Prefer to self-describe: "Gender queer" | 1 |
| Prefer to self-describe: "Uses male pronouns" | 1 |
| **Age range** |  |
| 18-30 | 9 |
| 31-50 | 6 |
| 51-70 | 3 |
| **Ethnicity** |  |
| White | 14 |
| Asian or Asian British | – |
| Black, Black British, Caribbean or African | 3 |
| Mixed or multiple ethnic groups | 1 |
| Other ethnic group | – |
| **FND Symptoms** |  |
| Functional seizures/Functional motor symptoms | 7 |
| Functional seizures/functional cognitive symptoms | 3 |
| Functional motor symptoms | 5 |
| Functional motor symptoms/functional cognitive symptoms | 3 |

"I do have a habit of trying to make things be less... I'm like *ohh, it's not as bad as it is, or it's not as serious or as important, or a big impact as it is.* Like, so when I first started FND I was like *it's not a big deal*." M1029 (EMDR)

Another participant shared that they attributed the experiences to themselves:

"And so, you can see this picture suddenly start to develop as to, I've probably had it for 20 odd years, but nothing really put a finger on any of it. I just put it to idiosyncrasies of me, but now we know we can say, probably at least 20 years." M1007 (NPC)

A few of the participants commented that they had a gut feeling and knew that something strange was happening with them, which was worth investigating. When prompted by the interviewer (BJ) about what they thought about their first FND symptoms, participant M1002 answered that:

"Worrying for sure, but, ah, for a while at least, I'd been content enough with the explanation of it being anxiety. It was only until like, you realise, oh, I really don't have control over my body, or these memory lapses are longer than I think." M1002 (EMDR)

In a similar vein, another participant told us:

"*No, no, no, you just need to look after yourself* and it's like, well, I mean yes, but also this feels like there's something more here." M1032 (EMDR)

**Table 2. Themes and sub-themes.**

| |
|---|
| **1. Is FND real?** |
| 1.1 Experiences of the onset of NFD |
| 1.2 "Why did this have to be something that nobody knows?" |
| 1.3 "None of you listened." – was this really the case? |
| 1.4 "Just get on with it." |
| **2. "This monster has a name."– a precarious path to diagnosis** |
| 2.1 Misdiagnosis |
| 2.2 Does a diagnosis always feel like the "sun coming out"? |
| **3. "It's FND, nothing can be done about it."** |
| 3.1 Discharge after diagnosis |
| 3.2 A dearth of treatment options |
| 3.3 "Here's two websites, good luck." |
| **4. Accepting FND – is it a "life sentence"?** |
| 4.1 "Pandora's box of symptoms" |
| 4.2 "Being wrapped in bubble wrap" – life impact |
| **5. "This isn't me controlling it… I don't like that very much."** |
| 5.1 "I can't remember what's me and what's not." |
| 5.2 "Trying to take control of it myself" |
| **6. Hopes for the future** |

**1.2. "Why did this have to be something that nobody knows?".** Almost all participants expressed how FND was something that nobody around them, including themselves, HCPs and people in their inner circle and social life, knew about.

"Yeah, but I mean, even at work, it's interesting because one of our board members was, I work from home a lot more than I used to, and she didn't like that until last year her niece was diagnosed with FND and currently can't get out of bed, can't even swallow properly, all sorts of things, and suddenly she's revised her thoughts on it, you know, because people don't think it's real until they know somebody with it." M1040 (EMDR)

As a first point of contact, many participants reached out to their GP for a consultation. Some, however, due to the nature of their symptoms, were initially assessed by emergency department clinicians of specialties other than neurology.

"The other GP is like *we don't, we don't know what this is, we don't understand it, we're not going to take it into consideration.*" M1021 (EMDR)

"Paramedics, they were doing their best, but they said *well, I've never seen this before, so I don't know how to deal with it.*" M1001 (EMDR)

"I went to neurology at my local hospital and cardiology because they were concerned, because I was, by that point had started having absent seizures and we were like a little bit concerned if it was like a head thing or a heart thing or I'm *just having a moment* thing, like what's going on. Umm, they kind of looked at what they could and did what they could, and but none of us really knew what was going on." M1026 (NPC)

The majority of the participants reported that they felt accused by HCPs of faking their symptoms, consequently leading to a few participants further questioning their own sanity.

"I did start going - am I mad, am I unaware that I'm like actually faking this without realising?" M1029 (EMDR)

"[HCPs] did lots of scans, said *there's nothing wrong with you, it's all in your head, go away, I think you might be a bit depressed or putting it on*." M1040 (EMDR)

"You could be in there, in the interview with your tremor playing up and they'll be like *that doesn't exist or are you putting that on*, because no doctor has put on a bit of paper that this is a thing." M1025 (NPC)

The lack of awareness and knowledge demonstrated by HCPs led to participants having to "play the expert around the people who should be the expert" (M1040).

"I actually said to them *could this be functional neurological disorder?* and I sent, I gave them a print out from FND Hope and they were like *no, no, no, it's just your anxiety*." M1026 (EMDR)

The journey of figuring out what is going on with their bodies and receiving uncertainty from those around them led to participants reporting negative emotions, such as loneliness, frustration, sadness, anxiety and more.

"I just think *oh, here we go again, I've gotta explain it, or get Google out* and also some people were like *ohh well, we're not sure that's really real* and you think *oh, just shut up*, so yeah, I'll say frustrated." M1040 (EMDR)

"When you have something that people don't know, you really are, you feel lonely anyway, but you feel even more lonely." M1014 (EMDR)

**1.3. "None of you listened."** – **was this really the case?.** Most participants recounted that they were subject to rude, dismissive attitudes from the treating healthcare specialists.

"I was in there the other week, I was having a seizure and one of the doctors walked in and said to my husband *is she going home* and he went *does it look like it?* Instead of apologising, he just walked away." M1007 (NPC)

"I think it didn't help that I had one doctor who said *it might be a tumour* and I just looked at him, he just said that really calmly and then left, like you just brought up the idea of a tumour and then just change and just moved on from it and I'm just sitting there, going like, *you can't just say something that's quite a big thing and then just leave it*, you know?" M1029 (EMDR)

"All of my experiences running up to having the diagnosis were very dismissive on my GP's end." M1012 (NPC)

However, some participants acknowledged positive interactions with both people from their inner circle and HCPs. These interactions were classified to be of supportive nature, where the participants felt heard, understood and accepted.

"I had a seizure at work, but the support that my colleagues have given me has been absolutely wonderful and I never expected that they would care about me as much as they have shown me to." M1032 (EMDR)

Only a few participants disclosed that they met HCPs who paid attention and/or knew of FND, such as a hospital consultant, GP, or neuro-physiotherapist. However, this was a rare occurrence, and the clinician had to be senior or to be curious about unexplained symptoms and eager to educate themselves on the matter. Participants usually described this instance as having the luck to meet and be under the care of HCPs who know.

"My main GP I see, she's fully conversant with it and understands it and she's superbly supportive…. because she's actually researched FND herself, after me sending her a leaflet. She has also done a training session with all the other practitioners at the surgery, so they will have someone understanding." M1007 (NPC)

"It's only when I spoke to one doctor in particular that he was, like, really interested and he was quite kind of like intent on working out how the different things connected. And he was the first person to listen." M1021 (EMDR)

**1.4. "Just get on with it.".** Post initial assessments, nearly all participants were left without management, correct established or hypothesised diagnosis, or follow-up, whilst waiting for specialist assessments.

"My GPs were also a bit just like, *yeah, you'll be fine, like, you always work this out.*" M1030 (NPC)

"At this point I didn't know it was FND, and I called up relaying my symptoms again and saying *ohh, like I physically can't walk in the mornings, it's really painful*, things like that and he was quite harsh and said *oh well, we've already had an MRI and there's nothing wrong with you, so just get on with it.*" M1012 (NPC)

**Theme 2. "This monster has a name."** – a precarious path to diagnosis

Following referral to specialist services, participants were given a diagnosis of FND, usually by a neurologist or neuropsychiatrist. The entire diagnosis process, from symptom onset until the FND label, took a long time, ranging from one year to twelve years, and involved ruling out other disorders and having multiple scans and tests.

"So I was with [neurologist at A&E] and he did a lot of tests and did the, I can't remember it's, but it's the one where you lift your leg up and he was like, oh yeah, this is FND, but I can't diagnose you with FND, you have to wait a year, I think it was eleven months for the referral to get diagnosed." M1024 (NPC)

**2.1. Misdiagnosis.** As part of the journey of arriving at the correct diagnosis, nearly all participants were misdiagnosed during their initial assessments with disorders such as epilepsy, multiple sclerosis, stress, anxiety, carpal tunnel syndrome, migraine, stroke and more.

"I think the only frustration that I had was trying to get the GP to kind of take it seriously, because I think they kind of just assumed *ohh, you know, you're just stressed, you need to look after yourself a little bit more.*" M1032 (EMDR)

"They really were of the mindset that this looks like textbook multiple sclerosis. *You're the right age, these are the initial symptoms of it*; they obviously said *we're not gonna tell you got that, but this is probably what we think you should prepare for.*" M1040 (EMDR)

**2.2. Does a diagnosis always feel like the "sun coming out"?.** Participants reacted differently to learning what FND is and receiving the diagnostic label. Some participants were content, and the rest reported feeling negatively impacted or experiencing mixed emotions.

"It was like a weight had been lifted off me. It was like the sun coming out, *oh wow, that's what it is* and you know, everything fits in there." M1016 (NPC)

"It felt a bit, it was nice just to have the bit of paper I think. And nice for someone to validate that I wasn't making it up and that this was actually a thing now." M1024 (NPC).

"*This is FND*. I was like *OK, I don't really know what to kind of do with that*, it was quite frightening." M1026 (NPC)

"It was such a mixed bag of emotions cause I was ecstatic to finally have an answer like reading on everything you're like *holy hell, that makes so much sense. That is me on a website*. Umm and yeah, so there's that finally getting some clarity about yourself. But then also it felt like a life sentence. It felt like *ohh, this isn't just some cute little quirky teen thing that's going to pass. You're telling me I'm stuck feeling like sh\*t? Ohh!*" M1002 (EMDR)

Even if some participants now had a name for their condition, in general the explanations by HCPs were simple and unsatisfactory, still leaving ambiguity and confusion.

"To be told there's a disconnect and it's sort of quite a cerebral thing if you like, I couldn't really get to grips with that and I couldn't understand where the end of the tunnel would be because it just seemed to be such a nebulous subject. I had great difficulty, as I say, getting to grips with that." M1011 (EMDR)

**Theme 3. "It's FND, nothing can be done about it."**

This theme illustrates participants' different experiences of treatment, or lack of options for treatment. Some participants had not been offered any treatment. Amongst those that were, the most common options reported were physiotherapy, medication, pain clinics and psychological therapy (such as cognitive behavioural therapy (CBT) or mindfulness).

**3.1. Discharge after diagnosis.** Across some cases, health professionals offered no plan or route forward after establishing a diagnosis, and patients were discharged after diagnosis.

"Kind of didn't expect to be discharged kind of then and there, I thought it was gonna be, they kind of said to me that it would kind of be like a review kind of every six months. So yeah, I found that quite hard because I did kind of just felt like I was kind of just out on my own again and kind of was unsure that, you know, if my symptoms did get worse, what I have to do." M1035 (EMDR)

"He [neuropsychiatrist] ticked off that box and I'm left in the same position I was in without any further support, so that is not nice at all." M1021 (EMDR)

**3.2. A dearth of treatment options.** The participants who were offered treatment reported that they faced long waiting lists and/or being transferred between different teams, and a few of them had to advocate for themselves to gain access to treatment in the first place. They all shared that the treatment options were not efficacious in helping their symptoms for both physical and mental difficulties. Some participants were referred to MODIFI as an alternative.

"The common theme here is things have to hit the absolute limit, you have to hit your last nerve and get so angry or end up having a very public, embarrassing breakdown of some kind and then you will get like a sliver of help." M1002 (EMDR)

"They just don't really know what to do or don't really understand. …possibly one of my favourite things I've heard from a doctor...So with my absent seizures…I've had a really lovely neuropsychiatry doctor tell me *when you feel a seizure coming*, which I don't always feel coming, *start doing grounding techniques*. Like I'm sure that's really helpful for some of your patients, but considering I can't feel them coming and I'm not aware during them, I can't really ground myself." M1026 (NPC)

"The only treatment idea I suppose I've had was I was put in this trial." M1026 (NPC)

**3.3. "Here's two websites, good luck." M1026 (NPC).** A few participants shared that they were told to 'self-treat' by using the guidance from the internet which is curated for FND. This lack of appropriate clinical management felt like there was no way out of the FND journey.

"Even at this point, I've been sent like the website. It does feel annoying though after like five years of misdiagnosis that you get to the end of it and it's like *here's two websites, good luck*." M1002 (EMDR)

"I can remember coming out of there where basically the treatment was, well, *here's some websites, go and learn about it* and I don't wish to be disrespectful to the doctor, but I really came away thinking *well, if I read up on it, that's as far as I'm gonna get*." M1011 (EMDR)

### Theme 4. Accepting FND – is it a "life sentence"?

When participants acquiesced to the fact that there is no current cure to FND and that they had exhausted the options for treatment and help from clinicians, they learned to navigate life with debilitating long-term symptoms and to accept the new reality.

**4.1. "Pandora's box of symptoms".** All participants' accounts revealed that with time, more symptoms appeared and became more volatile and frequent. Long-term symptoms included fatigue, panic attacks, tremors, and others.

"All the symptoms don't all happen in one go, it's, you know, one flares up and the other one dies down or something like that." M1016 (NPC)

"You know, it's how things are. You just have to continue with everything as it comes now. You take it day, you know for me, I live day by day, some days is hour by hour. Depending on whether it's a good or bad day. I know some days where, if I've been doing a lot or gone out and socialised, for the next two or three days I'm exhausted. But you know, it's fine." M1007 (NPC)

Coming to terms with the negative experiences of seeking a diagnosis and trying out treatment pathways, the majority of the participants expressed negative emotions, such as despair, guilt, worry, fear and anger.

"Think I have made my flare ups worse because I haven't let myself recover because I've been angry at myself for being unwell." M1021 (EMDR)

"But, uh, I'm still left with the feeling that I'm not gonna get, I'm never gonna lose this." M1011 (EMDR)

**4.2. "Being wrapped in bubble wrap" – life impact.** All the interviews portrayed the massive imprint that FND was leaving on a daily basis in participants' lives. Participants shared that they were unable to fulfil their household duties, go to work or school, play sports, engage in hobbies, drive and take part in other activities.

"I couldn't shower on my own, I couldn't cook on my own. It was, yeah, it was awful." M1035 (EMDR)

"I couldn't do up zips on jackets and things like that ….the tremor was just continuing, continuing, continuing. I couldn't write my name anymore." M1024 (NPC)

Participants adapted to a new way of living, primarily home-based, which allowed them to implement safety strategies aimed at minimising potential challenges associated with FND.

"It obviously, you know, it turned my world upside down. You know, one minute I was, you know, going out on my own and working and then the next minute I was, you know, stuck in the house, stuck in one room in case, you know, I fell and hit my head." M1035 (EMDR)

Consequently, being confined to the home environment increased participants' dependence on their families and disrupted existing relational dynamics. Participants reported that this imposed lifestyle—neither chosen nor desired—resulted in a significant loss of independence.

"My whole independence was taken away from me because for safeguarding reasons the school couldn't allow me to travel home on my own, so my dad had to pick me up. So all of my independence that with my autism I worked so hard to get, was just taken away from me like overnight. And so that was very annoying." M1029 (EMDR)

"I don't want to be that burden, I don't want to, I'm someone who's very independent and I don't want to be dependent…. I felt weak, I felt small." M1003 (EMDR)

Participants experienced the loss of their pre-FND lives and were compelled to make sacrifices and compromises to safeguard their physical health, often at the expense of their emotional wellbeing.

"Ohh, yeah, like *she's fine now, like she's having less seizures and she's having less motor symptoms and she's having less of that.* It's because I've had to give up so much and I have to consider so much." M1021 (EMDR)

"Everyone was going out and I couldn't because I was so exhausted, you know, from having kind of back-to-back seizures and it kind of just felt like my life had stopped, but everything else was carrying on." M1035 (EMDR)

**Theme 5. "This isn't me controlling it... I don't like that very much."**

Many participants expressed a sense of helplessness, referring to FND as 'this thing' over which they had no personal control. This perceived loss of agency over their bodies and lives elicited feelings of anger and intensified a strong desire to regain autonomy and return to a sense of normalcy.

"And I think, I think to be honest, I do have a bit of an ableist mentality because that's how I've been brought up. So, when I found out that I have this, I was like *no, I absolutely will not be limited by it*." M1021 (EMDR)

**5.1. "I can't remember what's me and what's not.".** Most participants conveyed losing a sense of their own selves, because FND created a new version of themselves and one which engulfed their life uncontrollably.

"Uh, so I think because I have lost confidence because I used to be really confident, but I don't go out, [neuropsychiatrist] is desperately trying to get me to get out the house." M1016 (NPC)

"I can't remember what's me and what's not, in terms of where I start, where I stop and it starts…I hate saying it because I don't think a medical condition should define anyone, but it's very much a part of me." M1030 (NPC)

"I always had to kind of get people to stay away from me because I was scared to like see other people because, I mean, I'd like to think I'm quite a kind person and I did find it very difficult that I had to, this isn't me controlling it and it was sometimes quite violent and sometimes like I said rude things and I just didn't really like that very much." M1026 (NPC)

"I can't even play the piano because I can't move my fingers like that, which considering art and playing the piano were very much who I was, it really took a toll on my mental health, which was already in a rubbish place anyway." M1030 (NPC)

**5.2. "Trying to take control of it myself".** This sub-theme reflects the yearning of all participants to get their "life a bit more back to normal (M1040)". Wishing things were different, some of the participants demonstrated proactivity in searching for answers, or/and helping others and not giving up.

"And I think a lot of it kind of was done actually by me and like looking into things and trying to take control of it myself, because I've only seen the neuropsychiatrist a couple of times and I'd already kind of done a lot of work before I'd seen him." M1021 (EMDR)

"You know, I'm kind of using my experiences to help others. I'm part of FND Dimensions now, I've been doing working with them. And sort of they're brilliant with their knowledge and help as well, and now I'm kind of trying to use some of my knowledge that I've learnt to help others to point them in the right direction." M1007 (NPC)

Joining online support groups and connecting with other individuals living with FND was reported to be both useful and not so useful by different participants.

"I've, you know, been out on the Internet, on the digital world. And I found, you know, other people with FND to connect with and see kind of how their journeys are going which kind of gives me hope I suppose." M1026 (NPC)

"It's almost going back to when I joined the Facebook group and I kind of like tapped out because I was like, this isn't what I'm experiencing." M1030 (NPC)

### Theme 6. Hopes for the future

The final theme represents the light at the end of the tunnel, which came from within. Most participants developed some understanding, coping strategies and skills, although unfortunately, these were generally felt to be insufficient. In the context of frustration and despair, participants were left with hopes for more – a better future, a "cure", a return to a previous lifestyle, more treatment options, bespoke management, and more information about FND.

"I was just hoping for more, was hoping for more consistent treatment. I was hoping for someone else, a professional to be doing the job I'd half-heartedly been doing for 4.5 years working my way through this." M1002 (EMDR)

"…that there's more like alternatives and there's other ways that people can get help." M1017 (EMDR)

"I just need someone to help me, actually hear that and go *OK, let's do what we need to do with that*. Because it's just not out there - that support for FND - I don't think, not yet, at least." M1026 (NPC)

### Discussion

This qualitative study provides a narrative of participants' experiences of living with FND from the onset of symptoms through seeking diagnosis to possibly accessing treatment. Six main themes were identified: (1) seeking a diagnosis and relief from the physical struggles, (2) receiving a diagnosis of FND, (3) treatment for FND, (4) the impact of FND on day-to-day life, (5) the need to take control of one's life and return to normal, and (6) hopes for the future.

The journey of FND began for our participants from the moment symptoms arose and help was sought. Where onset began in the early teenage years or in their twenties, some participants initially underestimated or ignored their symptoms. It seemed however that some participants experienced a 'gut feeling' that there was more to it and that symptoms warranted assessment.

Whilst some participants in this study sought direct assistance from primary healthcare providers, others were assessed by emergency services. A commonality between the different routes of assessment was that participants were passed between different medical disciplines following an unsuccessful attempt from clinicians to assess the ongoing issues and frequently received misdiagnoses. This would sometimes lead to needs being overlooked, or discharge with questions unanswered. Similar findings have been reported in a review of 127 studies [23,24], which highlighted that following a cascade of inadequate care, FND patients were left without treatment or monitoring.

The quest to obtain a diagnosis was challenging and frustrating for our participants. It was often unsuccessful due to the lack of knowledge from various HCPs, as has been reported in other studies [11,23,24]. Participants in this study also experienced negative attitudes from treating specialists, such as being dismissed and accused of feigning symptoms. According to research conducted in 2020 by the charity FND Hope, 82% of respondents worldwide reported feeling rejected or disbelieved by medical staff [25]. In line with reports from other studies [1,4,9,26,27], most HCPs and friends/work colleagues reportedly doubted the validity and possible severity of FND. This tallies with the stigma which commonly accompanies FND. Conversely, some participants reported that they encountered positive attitudes and compassionate care by HCPs and people in their lives.

Many participants in this study were disabled due to their FND symptoms, in line with previous studies on the impact of FND [28–32]. Individuals faced multiple issues, with limited mobility in daily life and difficulty completing activities that they would have done before FND onset, (e.g., driving a car, showering unassisted). Daily activities were reported to be impacted to the extent of having to leave school or stop working. Participants described how their social lives were put on pause, and it was common for them to choose to stay at home where they would feel safer. A description of dependence on others growing exponentially was common and some participants reported it affecting their self-esteem and causing a negative psychological impact. Similar to another study [29], participants felt a loss of control over their lives and their capabilities due to the nature of their FND symptoms.

Participants expressed the need to regain their sense of agency and to return to their old selves. Losing one's identity, as a result of living with FND, was a phenomenon that was expressed by the majority of the participants, and this sense of loss was accompanied by grief. They described that as FND consumed their lives, it was hard to maintain the roles of being a parent, partner, colleague, friend or family member, which has also been reported in the study of Dosanjh et al. [26]. Participants gained a new focus on life with an identity of somebody who is ill, separate from the roles they used to have in society [31]. Illness identity affects how one thinks about oneself [33]; in this study, the majority felt that FND dominated their lives, similarly observed in other chronic conditions like multiple sclerosis [27,28].

Receiving a diagnosis, often by specialists in the field of neurology, took a long period of time, as noted in wider literature [4,26]. Some participants felt confused and struggled with understanding what the diagnosis meant, others feared what the future would look like. Still, for many others receiving a diagnostic label brought a sense of relief [28,34] and reassurance in their own sanity due to doubts provoked by the disbelief of others [32].

After receiving a diagnosis, some participants were not offered any treatment, while others were offered options including physiotherapy and psychological evaluation or therapy. Despite the experience of different treatments, all participants reported that nothing had fully alleviated their FND symptoms and often referred to the treatment options offered as "unhelpful".

Holding out hope that their life could return back to normal and driven by despair and anger, some participants deemed it necessary to advocate for themselves and seek further support to manage their long-term symptoms. Engagement with their condition included doing online research through websites such as FND Hope, seeking private or international care and joining peer support groups on social media. In joining FND groups for their own benefit, they found that their experience could also benefit others, which gave them a sense of purpose by helping other individuals who were struggling with FND. Some participants found online networks to be a useful avenue, whilst others did not identify themselves with such groups' patient population due to their different struggles, similar to previous findings [28]. The long-term ambitions and hopes of many participants were to receive bespoke or tailored treatment, have a greater range of options for treatment, and more information – all emphasising the need for advancements in healthcare.

The findings from this study support multiple recommendations from the literature. Delayed diagnosis and inappropriate treatment can lead to a poorer prognosis in FND [4] and increase the cost of addressing FND, which is already reported as a high financial strain on the healthcare system [35]. In order to cater to patients' hopes for an earlier diagnosis, training targeting medical professionals' gap in knowledge regarding making an FND diagnosis may be a

solution. O'Keeffe et al. [4] highlighted that doctors often worried about treating FND patients, viewing them as difficult cases. Addressing the quality of education on FND in the medical sphere could lead to a decreased time to diagnosis, avoiding the need to be assessed by multiple services, and increasing service user experience. For example, a study from Medina et al. [36] demonstrated that a one-hour workshop led to HCPs reporting feeling more comfortable with assessing FND, understanding it and explaining it to patients, and also increasing belief in the validity of FND symptoms and having greater acceptance of FND patients. Mcloughlin et al. [37] synthesized evidence in a scoping review to reveal that current teaching about FND during medical training is insufficient, and change has only been observed very recently within the UK neurology training curricula. Additionally, as some participants reported feeling confused regarding their diagnosis, patient education could be enhanced, in addition to their medical appointments, e.g., patient education group sessions [38]. However, it is difficult for professionals to offer a diagnosis, but then not have a treatment pathway for patients. Therefore, in addition to improved diagnosis, expansion and improvement in multidisciplinary input for treatment of FND is required [1,39–42], in line with the National Neurosciences Advisory Group optimal care pathway and NHS England's Specialised Neurology Services Specification [43]. Lack of access to suitable treatment was a strong theme in our study and underpins calls for the development of accessible and coordinated clinical treatment pathways [44]. Additionally, greater public awareness of FND is also necessary in order to reduce stigma and enable people to access appropriate help earlier, particularly if they are not in contact with the medical system. This study is one of the very few in current international literature, and one of only a few UK-based studies which examines the experience of living with FND. We recommend that future research includes more qualitative studies to provide a platform for the voices of people with FND in research.

### Strengths and limitations

This study's main strength is the rigorous approach used to recruit participants and collect and analyse the data, adhering to quality and validity criteria of good qualitative research. Purposive sampling has included participants from MODIFI who are diverse in representation of different experiences. The sample size is sufficient to gain depth of people's individual FND experiences and carry information power. We also included lived experience experts throughout all stages of the research, from design to analysis and reporting.

There are also limitations to be noted. The researchers most closely involved in conducting this qualitative study were all white women, although the wider group involved brought more diversity to the team. The purpose of the interviews was also not to solely explore what it was like to live with FND; the interview was part of wider interviews investigating the experiences of participants on the MODIFI trial, and thus this trial participation may have altered patients' perspectives, potentially more positively, given the promising results of the trial. The findings of this study are consistent with broader international literature on the subject, but it is important to note that due to the nature of the trial, only participants in England were included, and our findings may be context-dependent to the UK.

### Conclusion

This qualitative study presents six key themes that reveal common, often negative experiences of individuals with FND. The findings emphasise the disabling impact of FND and how much the lives of participants changed as they experienced losses, and profound negative effects on their health and wellbeing. The challenge of living with FND was accentuated by the existing barriers of accessing treatment and help. Echoing similar recommendations from other studies, we suggest that future research is needed to address the gaps in knowledge about FND in order to refine clinical guidelines. We further suggest that support for this population could be advanced through different systemic and social interventions, such as multi-disciplinary treatment of FND patients and increased awareness of FND. It is crucial that more is done to improve service delivery within healthcare to produce a positive change in the quality of life for people living with FND.

 

## Supporting information

**S1 File. Interview schedule.**
(DOCX)

## Acknowledgments

We would like to extend our thanks to the participants who took part in the interviews.

## Author contributions

**Conceptualization:** Sarah R. Cope, Mark J. Edwards, Jared G. Smith, Sharif El-Leithy, Serena Vanzan, Patricia Hogwood, Dawn Golder, Kati Jane Turner.

**Formal analysis:** Ivet Pritomanova, Patricia Hogwood, Dawn Golder, Kati Jane Turner, Jo Billings.

**Funding acquisition:** Sarah R. Cope.

**Investigation:** Billie James.

**Methodology:** Sarah R. Cope, Billie James, Jo Billings.

**Project administration:** Serena Vanzan.

**Supervision:** Jo Billings.

**Writing – original draft:** Ivet Pritomanova.

**Writing – review & editing:** Sarah R. Cope, Billie James, Mark J. Edwards, Jared G. Smith, Sharif El-Leithy, Serena Vanzan, Patricia Hogwood, Dawn Golder, Kati Jane Turner, Jo Billings.

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
