## [Decision Letter · Decision Letter 0]

27 Oct 2025

Dear Dr. Billings,

Thank you for submitting your manuscript to PLOS ONE. After careful consideration, we feel that it has merit but does not fully meet PLOS ONE’s publication criteria as it currently stands. Therefore, we invite you to submit a revised version of the manuscript that addresses the points raised during the review process.

We look forward to receiving your revised manuscript.

Kind regards,

Ioannis Liampas, MD. PhD

Academic Editor

PLOS ONE

Journal Requirements:

“NIHR RfPB Grant Reference Number 202277”

4. In the online submission form, you indicated that [Due to the personal nature of participants' accounts, and in line with our ethical approvals, raw data from this qualitative study are not available publicly outside of the research team. Reasonable requests for further information about the dataset can be made to the corresponding author.].

Reviewers' comments:

Reviewer's Responses to Questions

**Comments to the Author**

1. Is the manuscript technically sound, and do the data support the conclusions?

Reviewer #1: Yes

Reviewer #2: Yes

Reviewer #3: Yes

2. Has the statistical analysis been performed appropriately and rigorously?

Reviewer #1: I Don't Know

Reviewer #2: N/A

Reviewer #3: N/A

3. Have the authors made all data underlying the findings in their manuscript fully available?

Reviewer #1: No

Reviewer #2: No

Reviewer #3: No

4. Is the manuscript presented in an intelligible fashion and written in standard English?

Reviewer #1: Yes

Reviewer #2: Yes

Reviewer #3: Yes

Reviewer #1: Thank you for this valuable work tacking such an important and neglected topi

In line 90 what PPI abbreviation for?

In line 124 " two preferred to self-describe and one preferred to not disclos" what does it mean?

In line 140 you mention fourteen subthemes while in table 2 there are only thirteen subthemes.

Reviewer #2: The authors present an interesting qualitative study, where the lived experience of an individual with FND is described. They conducted a series of interviews, and analysed the responses according to standard analytical pipelines. They extracted the relevant themes, and discuss them extensively.

The paper is overall well written, and original. To my knowledge, the lived experience of an FND patient is not well-described in the literature, and this study addresses this gap.

However, I have some comments that should be addressed before the paper can be accepted for publication.

Introduction.

The introduction provides a clear overview of FND and its challenges. It could be strengthened with a brief statement of FND prevalence or incidence to emphasize its commonality

Methods

The recruitment strategy could be better described. How were the initial patients selected? What effort were made to ensure that diverse perspectives were taken into account?

Sample characteristics. It would be great to describe the sample with more details. For instance, the type of symptoms (motor, PNES, mixed, etc) should be presented, as well as the age at the time of the interview. As the time to diagnosis, as well as the misdiagnosis, were important themes that emerged from the interviews, this information should be reported too. What was the mean time to diagnosis, and how many wrong diagnosis were being given before FND?

For a thorough methodological description, I would suggest to also add the mean duration of the interviews.

It would be helpful to provide, perhaps as supplementary material, the interview guide.

Results

Some quotes appear to be very long, and sometimes repetitive. I would suggest to keep only the quotes that describe different concepts.

Discussion

The discussion presents a detailed synthesis of the main results, and concludes that the health care professionals should receive better training in treating and managing FND. I wonder if more translational recommendations can be formulated, for instance, in which aspect of FND management the HCPs should receive better training.

The discussion on the workshop seems to diverge from the main message of the paper, and I suggest shrinking it.

I wonder if the results of this study may also point to training the affected individuals.

Reviewer #3: This is a beautifully written manuscript describing a qualitative study regarding the experience of those with Functional Neurological Disorder (FND), from first symptoms to the diagnosis and potential connection with treatment. The research team appears very well qualified to be conducting this study and has demonstrated a commitment to ethical and rigorous procedures. Their findings illuminate common themes identified in previous studies regarding the often frustrating experience of healthcare provider and public lack of understanding of FND, stigmatizing and accusations of faking symptoms, the incredible life changes that happen as a result of the diagnosis and its potentially debilitating symptoms, and the lack of hope for a symptom-free future.

The following comments or questions primarily point to areas where additional details could improve the reading experience for the audience and provide greater evidence of rigor. The comments/questions are organized by manuscript section.

Title: The title suggests the manuscript only covers details up to the diagnosis of FND; however, there are details regarding what happens after diagnosis in life and treatment as well. The title may benefit from being more reflective (such as The Journey to a diagnosis and care of FND?).

Abstract: Before arriving at the purpose, it may be helpful to provide a sentence or two about the background of FND. This will help support why the purpose of this study is significant.

Introduction: Line 59 notes a study regarding patients’ lived experience of FND (a systematic review of 12 papers). Then, in line 69, a statement regarding minimal investigation of the lived experiences of life with FND provides support for the need for this study. This study would be strengthened by a description of how this study is different from all the studies included in the systematic review about the lived experiences of FND. What is being added to science by this study?

Materials and Methods-Reflexivity: The authors provide a helpful explanation of reflexivity related to research team’s role, experience with FND. Because this is a reflexive thematic analysis, more detail related to the potential impact these roles and experiences may have on the analysis would be helpful. Were there any assumptions the researchers may have brought to the analysis or even included in the analysis as is allowed in reflexive thematic analysis.

Materials and Methods-Methodology:

• What did standard neuropsychiatric care (NPC) consist of in the parent study?

• For the interviews that were done in-person, where did these interviews occur?

• Including example interview questions in the text could help readers see how the questions may have influenced the organization of the data or the patterns and themes found.

• The authors provided great detail regarding the data analysis. The description may be supported by an exemplar of raw data and its progression from code to sub-theme to theme.

• Was there an audit trail used to enhance rigor or trustworthiness?

• How were interviews recorded?

• Who transcribed the interviews—team members or an external service?

Materials and Methods-Participants:

• How were the parent study participants approached and invited to participate in this qualitative study?

• What were the inclusion/exclusion criteria for the parent study? Were the additional inclusion/exclusion criteria for participation in the qualitative study?

• How many interviews were conducted via Teams and how many were in-person?

Materials and Methods-Ethics:

• Were there any safety concerns addressed during the study, such as symptoms/seizure occurring during an interview, statements of thoughts of suicide or severe mental/emotional distress?

• Were participants incentivized to participate in the qualitative study?

• What strategies were used for data security and protection of confidentiality (such as use of participant code numbers)?

Results:

• The themes and sub-themes table was very helpful, and the results were presented in a very clear, understandable format. It was very easy to follow. The exemplar quotes were very appropriate and clearly supported the themes and sub-themes identified.

• Line 208-I believe the H in FND Hope should be capitalized.

• Line 272-is there a word missing between nearly and participants?

• Line 306-308-This sentence was a bit confusing (primarily the use of the word few). Was the most common experience for participants NOT being offered any treatment at all? And then following that most common experience, the next most common experiences were being offered physiotherapy, medication, pain clinics, and psychological therapy (such as CBT or mindfulness)?

• Line 308-I believe this is the first, and only, use of CBT. Consider replacing CBT with cognitive behavioral therapy.

• Were there differences noted in responses based upon whether a participant was in the intervention or control group? This may go beyond the scope of this paper; however, I think other readers will wonder this same thing. Depending on what NPC included, could there be differences in study treatment experiences that may have influenced their perceptions of FND treatment?

Discussion:

• A thorough summary of findings was appropriately provided at the beginning of the discussion section. The paragraph starting at line 526 seems to unnecessarily repeat some results before reconnecting findings with extant literature.

• At the conclusion of the discussion, I was unsure what new and unique contribution this paper made to FND science. The authors point out that this is one of the only UK-based studies examining lived experience of FND. Are there reasons why the experience may be different in the UK compared to other countries?

Strengths and limitations: Lines 575-578-Readers may benefit from a bit more explanation as to how participation in the trial may alter patients’ perspectives related to life with FND, and, therefore, the findings from this qualitative study.

Conclusion: Line 592-What is meant by tailored comprehensive management of FND? This seems to be a new concept mentioned in the conclusion. Is there support for this recommendation in the discussion that could be connected to this conclusion statement?

I commend the research team on a thoughtful examination of the lived experience of FND, from symptoms to diagnosis and life beyond the diagnosis. This is important work that gives voice to the harrowing experience of living with this complex condition.

.

Reviewer #1: No

Reviewer #2: No

Reviewer #3: No

---

## [Decision Letter · Decision Letter 1]

2 Mar 2026

The journey to diagnosis and care of Functional Neurological Disorder (FND)

PONE-D-25-32316R1

Dear Dr. Billings,

We’re pleased to inform you that your manuscript has been judged scientifically suitable for publication and will be formally accepted for publication once it meets all outstanding technical requirements.

Kind regards,

Ioannis Liampas, MD. PhD

Academic Editor

PLOS One

Additional Editor Comments (optional):

Reviewers' comments:

Reviewer's Responses to Questions

**Comments to the Author**

Reviewer #1: All comments have been addressed

Reviewer #2: All comments have been addressed

2. Is the manuscript technically sound, and do the data support the conclusions?

Reviewer #1: Yes

Reviewer #2: Yes

3. Has the statistical analysis been performed appropriately and rigorously?

Reviewer #1: I Don't Know

Reviewer #2: N/A

4. Have the authors made all data underlying the findings in their manuscript fully available?

Reviewer #1: Yes

Reviewer #2: No

5. Is the manuscript presented in an intelligible fashion and written in standard English?

Reviewer #1: Yes

Reviewer #2: Yes

Reviewer #1: Thank you for this valuable work in such an important topic and thank for addressing all comments

Tha manuscript is well written and informative and presented in an intelligible manner.

Reviewer #2: Thank you for this revised version. All my comments have been addressed, and I recommend this study for publication.

.

Reviewer #1: No

Reviewer #2: No

---

## [Editor Report · Acceptance letter]

PONE-D-25-32316R1

PLOS One

Dear Dr. Billings,

I'm pleased to inform you that your manuscript has been deemed suitable for publication in PLOS One. Congratulations! Your manuscript is now being handed over to our production team.

Kind regards,

on behalf of

Dr. Ioannis Liampas

Academic Editor

PLOS One